# Serial attentional resource allocation during parallel feature value tracking

**Christian Merkel[1]\*, Luise Burgmann[1], Mandy Viktoria Bartsch[2], Mircea Ariel Schoenfeld[1,2,3], Jens-Max Hopf[1,2]**

[1]Department of Neurology, Otto-von-Guericke University, Magdeburg, Germany; [2]Department of Behavioral Neurology, Leibnitz Institute for Neurobiology, Magdeburg, Germany; [3]Kliniken Schmieder, Heidelberg, Germany

**Abstract** The visual system has evolved the ability to track features like color and orientation in parallel. This property aligns with the specialization of processing these feature dimensions in the visual cortex. But what if we ask to track changing feature-values within the same feature dimension? Parallel tracking would then have to share the same cortical representation, which would set strong limitations on tracking performance. We address this question by measuring the precision of color representations when human observers track the color of two superimposed dot clouds that simultaneously change color along independent trajectories in color-space. We find that tracking precision is highly imbalanced between streams and that tracking precision changes over time by alternating between streams at a rate of ~1 Hz. These observations suggest that, while parallel color tracking is possible, it is highly limited, essentially allowing for only one color-stream to be tracked with precision at a given time.

## Editor's evaluation

This study presents an important finding on how human observers keep track of continuously changing feature values across two different streams but within the same dimension. The conclusion about the serial attentional resource allocation during parallel feature value tracking is informative to understand the function of visual cortical systems. The experimental evidence supporting the conclusion is convincing.

\*For correspondence:
christian.merkel@med.ovgu.de

**Competing interest:** The authors declare that no competing interests exist.

## Introduction

The visual system is very proficient in maintaining information about multiple objects at different locations over time. One common assumption to explain this is a central processing resource that is flexibly distributed amongst multiple relevant locations. Such spatial selection mechanism also enables the formation of parallel mental representations of several visual features at distinct locations through visual working memory (*Luck and Vogel, 1997*; *Scolari et al., 2008*; *Fukuda et al., 2010*; *Merkel et al., 2021*). Furthermore, in a dynamic setting, multiple relevant items can be tracked through retinal space over extended periods of time by enhancing the representations of individual spatial locations simultaneously (*Alvarez and Franconeri, 2007*; *Cavanagh and Alvarez, 2005*; *Merkel et al., 2014*). The actual distribution of the central processing resource amongst the relevant locations is hereby internally adjusted by task properties that enhance or reduce interference within the relevant, location-based, domain (*Shim et al., 2008*; *Franconeri et al., 2010*; *Chen et al., 2013*).

Interestingly, it has been suggested that the above-mentioned attentional tracking processes may not solely be tied to visual information traveling through space. *Blaser et al., 2000* showed that a visual object can be tracked through feature-space, in that separate visual features can follow

independent trajectories despite them being spatially inseparable. In other words, subjects are able to track an object through a simultaneous change in three separate feature dimensions. This finding supports the notion of object-based attention, according to which attention modulates the processing within all relevant features dimensions the attended object is comprised of *Duncan et al., 1997*; *Schoenfeld et al., 2014*.

Recent data suggests, that the visual system may also be able to maintain multiple changing feature values within a single feature dimension (*Re et al., 2019*). For example, *Re et al., 2019* asked subjects to continuously attend two superimposed color dot-motion clouds and detect faint saturation changes in either of them. The time-course of response accuracies to detect changes of the target colors was taken to suggest an oscillatory mechanism that allocates feature-based resources sequentially to all relevant items within the feature dimension (*Re et al., 2019*). Such rhythmic sampling of visual information to overcome resource limitations has been discussed before extensively as a mechanistic framework for spatial attention (*Landau and Fries, 2012*; *Fiebelkorn et al., 2013*; *VanRullen, 2016*), and it has been linked to oscillatory processes measured with electrophysiological recordings (*Busch et al., 2009*; *Song et al., 2014*; *Landau et al., 2015*; *Kienitz et al., 2018*; *Fiebelkorn et al., 2018*).

However, all previous research investigating multiple feature tracking addressed the issue by asking observers to track constant feature-values (e.g. two superimposed colors) to detect target changes in another feature dimension (e.g. a change in luminance or saturation). Importantly, this form of feature tracking does not require to continuously update the representation of changing feature-values within the attended streams. Hence, the question whether feature-based processes are able to maintain representations of multiple changing feature-values within the same dimension, in analogy to multiple-object tracking in space (*Alvarez and Franconeri, 2007*; *Schoenfeld et al., 2014*; *Pylyshyn, 1989*), remains entirely unresolved.

Here, we report experiments designed to directly assess the process of feature-based resource allocation during a highly controlled multiple color tracking task, where subjects are asked to simultaneously track changing but spatially inseparable color streams. Specifically, observers track the color of two superimposed dot clouds (*Figure 1a*) that simultaneously move along independent trajectories through color-space. Color-stream specific resource allocation is quantified as the precision of reporting one or both of the tracked colors. Cognitive resource distribution indexed by the proportion of precision estimates is measured across pairs of simultaneously tracked color-streams (experiments 1 and 2) and across variable tracking intervals (experiments 3 a-c).

## Results

### Experiment 1

Subjects were asked to simultaneously track the color change of two donut-shaped superimposed dot-clouds as illustrated in *Figure 1a*. The calculated trajectories of the two dot-clouds through color space were kept random and unpredictable, while never falling below a minimum critical distance, on each experimental trial, and the stimuli were designed in a manner that tracking could not be based on spatial information (c.f. Materials and methods for details). Color tracking lasted for 6–8 sec, after which both dot-clouds turned into grey. This prompted subjects to report the last perceived color of each stream (Target a, Target b; before turning grey), by dialing in the color as precisely as possible on a color wheel using a rotary knob (*Figure 1b* – Exp1). The color wheel was presented twice requiring the subjects to dial in the colors of both streams in sequence (Response 1, Response 2). The precision of the representation for the two color-streams (Precision High, Precision Low) was derived from the two standard deviations of a mixture model of two von Mises distributions containing pairs of target-response distances for all trials. On a given trial, the target (a or b) with the smaller of the two possible target-response distances (min(response 1 – target a, response 1 – target b)) was assigned to response 1 assuming a generally higher confidence for the first given response. The remaining target-response pair was assigned to response 2 (*Figure 2a*). Despite the fact that the resulting absolute precision estimates varied considerably across subjects (range(p1)=9.489, range(p2)=30.995) (*Figure 3a*), corresponding precision estimates of both streams showed a similar relation across subjects with a correlation of r=0.602 (p=0.008). A regression model with zero intercept (p1=a*p2) yielded a significant slope of a=1.748 (F(1,17) = 10.588, p=0.0046), suggesting an unbalanced resource distribution between the two streams of a similar ratio in each subject, independent of the overall precision of the

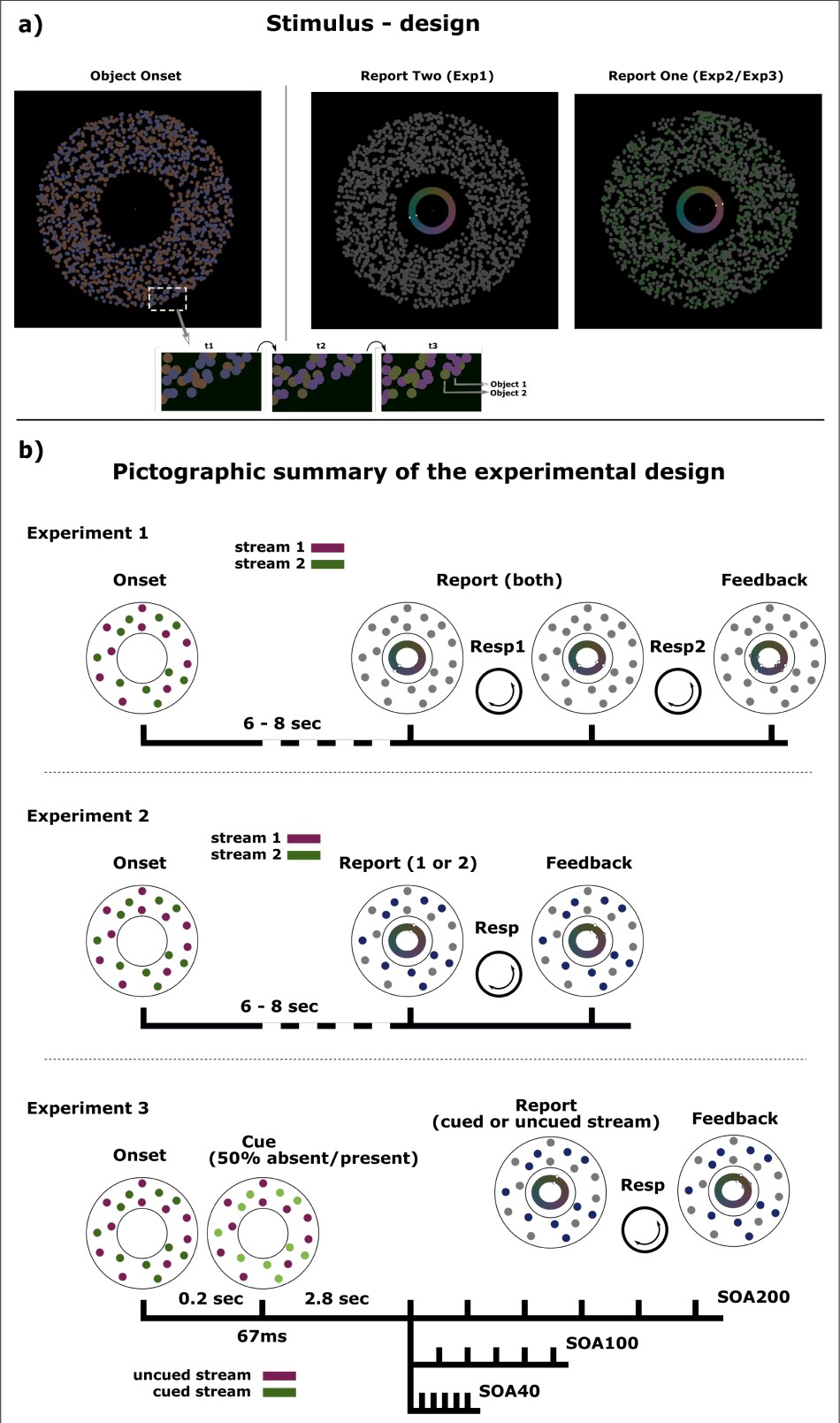

**Figure 1.** Experimental paradigm. (**a**) General stimulus design: Two superimposed dot clouds are present throughout each trial. Subjects are asked to attend to the colors of both objects as they transit through hue-space. In order to remove any spatial depth cue specific to one of the two objects, individual dots are superimposed randomly at time of initial generation. Furthermore, 10 random dots of each objects interchange identity (current

*Figure 1 continued on next page*

*Figure 1 continued*

color) at each frame throughout the trial, thus discouraging the generation of any spatial configuration in aiding feature tracking. At the end of the tracking phase one (Exp2/Exp3) or two (Exp1) of the streams change to grey and subjects have to report the last perceived color for that object using a dial. (**b**) Experimental design and time-course of trials for all three experiments. In experiment 1 and 2, subjects are tracking both color streams for 6–8 sec after which they are asked to report back on one (Exp1) or two (Exp2) of the streams. Probed color streams turn grey to indicate that they have to be reported on. After dialing in the response with a cursor on a centrally presented color wheel, subjects receive feedback about the correct response via a second cursor on the same color wheel. Experiment 3 additionally introduces a luminance cue for one of the two streams at the beginning of the tracking phase in 50% of the trials. After a set of fixed tracking intervals, subjects have to report on the color of the previously cued or uncued stream. This experiment is performed with three different sets (SOA200/SOA100/ SOA40) of six tracking intervals.

subject. Importantly, none of the pairs of precision estimates were different from a Monte-Carlo simulated set of responses in any of the subjects (p>0.811), which indicates that the variation in precision between streams arises from independent distributions. However, subjects were to report the color of the two streams in sequence. This may have caused recency effects, that is, the representation of the later reported color may decay while giving the first response. Accordingly, the difference of precision estimates may not reflect an imbalance of resource allocation during color tracking but a recency-based imbalance generated in the response phase of the experimental trial. Experiment 2 addresses this possibility, by having sixteen subjects of experiment 1 track two color streams exactly as in experiment 1, but report only one color-stream on a given trial (*Figure 1b* – Exp2). The logic behind this manipulation is that the precision measures of just one color will still sample from both color-streams randomly, but remove the ambiguity of two possible target-response pairings for each trial. The precision estimates will therefore represent a mixture of distributions. A difference between precision distributions as seen in experiment 1, would verify that the imbalance of resource allocation is truly arising in the color tracking phase.

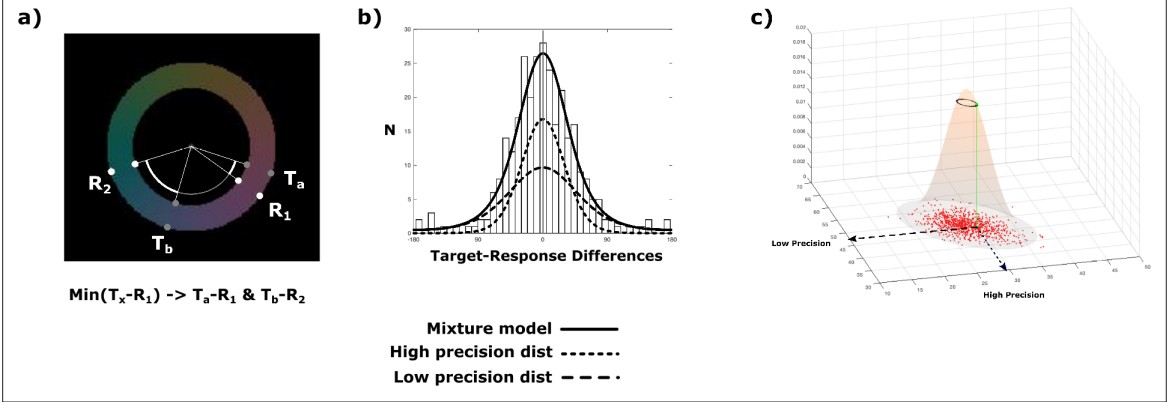

**Figure 2.** Aquisition of precision responses per trial and mixture modeling. (**a**) In experiment 1, subjects have to report on two target colors in sequence. The two reports ($R_1$ and $R_2$) and the two target colors ($T_a$ and $T_b$) for one trial create two possible target-response associations (and therefore two pairs of precision responses): ($T_a$-$R_1$ & $T_b$-$R_2$) or ($T_b$-$R_1$ & $T_a$-$R_2$). This ambiguity was resolved by pairing targets and responses based on the minimum angular difference for the first response to one of the targets (Min($T_x$-$R_1$)), and the second response to the remaining target ($T_{-x}$-$R_2$). This pairing is based on the assumption, that the first response would always be performed with higher confidence by the subject. It would be highly unlikely that for one trial subjects' first response would be a guess and the second response an accurate report based on correct tracking if they were able to only track one stream. Additionally, this leaves the possibility of the second response still being more accurate than the first response (by chance or actual cognitive performance). (**b**) Von Mises mixture model utilized in the experiments. The distribution of all precision responses (angular differences between response and target) would be the sum of responding each of the two tracked streams with each being allocated a certain cognitive resource (high/low precision). (**c**) The results of the Von Mises mixture models (as the pair of precision responses) is tested by Monte-Carlo-simulations. We tested, whether the parameters of the Von Mises mixture models can be estimated in artificially simulated sets of high and low precision target-response distributions based on those same parameters. The simulations confirm, that the estimated parameters reflect the ground truth of a sum of two precision distributions and not just a general inherent property of the model.

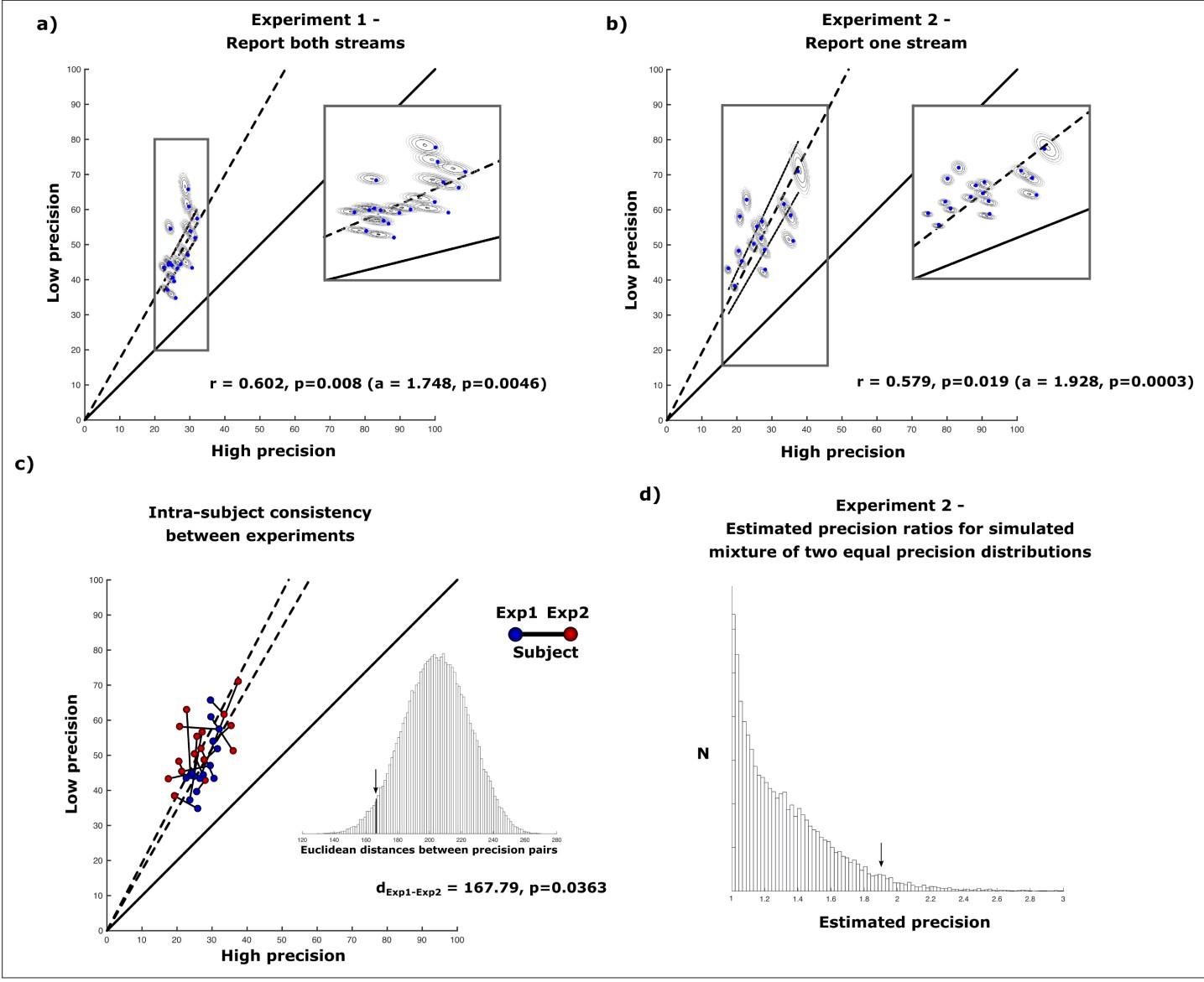

**Figure 3.** Results of uneven attentional resource distribution amongst two parallel color streams in experiment 1 and 2. (**a**) High and low precision estimates for each of the tested subjects (n = 18) (blue dots) in experiment 1. Additionally, the results of the Monte-Carlo simulated precision estimates for each subject are displayed as two-dimensional gaussians. In general, subjects are almost twice as precise in responding to one of the streams compared to the other stream. (**b**) Results for experiment 2. When subjects (n = 16) have to report only one stream per trial (while still attending to both color streams), the general ratio of high and low precision responses remain. (**c**) The same subjects show very similar pairs of precision estimates in both experiments (irrespective of whether they have to respond to both streams sequentially or only one stream). (**d**) Estimated precision estimates for mixture models containing two distributions with equal deviation based on the means of each subjects estimated precisions. The estimated precision ratio of Mixture models containing two equal distributions reaches a median ratio of 1.24 and values higher than 1.928 in less than 5% of conducted simulations.

## Experiment 2

The precision measures were subjected to a von Mises mixture model testing two distributions. Monte-Carlo estimates confirm that they do not differ from simulated datasets in any of the subjects ($p > 0.972$). Furthermore, the pairs of high and low precision values are found to be correlated across subjects ($r = 0.579$, $p = 0.019$). As in experiment 1, the relation between high and low precision estimates was analyzed via linear regression with zero intercept, which yielded a significant slope of $a = 1.928$ ($F(1,15) = 21.84$, $p = 0.0003$; *Figure 3b*). The slope of slightly below 2 is similar to the one obtained

when testing both color-streams in experiment 1, suggesting a robust subject-specific feature-based resource allocation that varies little across tasks.

In order to test computational biases of the mixture model itself potentially producing high ratios between the two precision estimates, a number of target-response distributions were created using von-Mises mixture distributions based on precision responses with equal deviation. Those distributions were based on the estimated deviations for each subject by taking the mean of the high and low precision estimates. *Figure 3d* illustrates a histogram of estimated ratios for 1000 permutations of our model if the true ratio between precisions, that is the slope, would be equal to 1. With a median of 1.24 less than 5% of the estimated ratios reach a value larger than 1.928.

Because all but two of the subjects of experiment 1 took also part in experiment 2, the difference in precision estimates between streams could be tested for consistency across experiments. The consistency was quantified by the Euclidean distances between precision estimates in experiment 1 and 2 (s=167.79) spanning a two-dimensional precision space containing high- and low precision measures. Specifically, for each subject, we determined the distance between corresponding points of high and low precision values in experiment 1 and experiment 2 (precision pairs). The distances were then summed and tested against the sums of 10,000 random permutations of the estimated precision pairs among subjects. The measured distance between experiments was significantly smaller than the distance obtained from random permutations (p=0.0363; *Figure 3c*), indicating that the ratio of precision estimates is very consistent within the same subjects across experiments.

## Experiments 3a-c

The ~2:1 ratio of precision estimates for the two color-streams suggests a strongly limited resource that entails a trade-off assignment during tracking. This could arise because subjects consistently devote more resources to one stream than the other. Another possibility is that over time the trade-off allocation changes in a dynamic fashion randomly or periodically between the attended feature values. Experiments 1 and 2 do not allow to distinguish among these possibilities, because which stream is actually attended at a given moment remains undefined, as we just probe at a random time point after the onset of the color-streams.

Experiments 3 a-c were designed to allow us to systematically assess the temporal dynamics of the attentional allocation process among the attended color streams. To this end, two critical modifications of the experimental design were introduced (*Figure 1b* – Exp3): (1) At the beginning of half of the trials the brightness of one dot group was transiently enhanced (cue-present trials), to bias subjects to attend this stream first and therefore temporally reset any time-varying process involved in resource allocation towards that stream. This ultimately enabled us to identify the cued and uncued stream at the point of report. (2) The precision estimates were systematically sampled at six fixed SOA increments starting at 2.8 sec after the brightness cue. The actual time-course of concurrent resource allocation during double color tracking is unknown yet. In order to gauge change rate optimally, we used three SOA versions (a-c) of the experiment, each run in a different group of participants (n=15). Specifically, SOA increments of 200ms, 100ms, and 40ms were run in experiment 3 a, 3b, and 3 c, respectively.

*Figure 4a* displays the pairs of low and high precision estimates for cue-present (red) and cue-absent trials (blue) of each subject separately for experiment 3 a-c. Correlations and regressions between low and high precision estimates were calculated excluding three, three and two visible outliers in each of the experiments, respectively. For experiment 3 a (SOA200), low and high precision estimates were correlated for the cue-present trials (*r*=0.5802; p=0.048) as well as the cue-absent trials (*r*=0.6617; p=0.0191). The slope of these relations was similar for both factor levels (cue-present: a=1.8205 ($F_{(1,11)}$ = 13.124; p=0.004) | cue-absent: a=1.9145 ($F_{(1,11)}$ = 15.328; p=0.0024)). Hereby, using cued and uncued trials to derive high and low precision estimates yields highly similar results (d=60.6866, p<0.001). In experiment 3b (SOA100), the pattern of results was comparable. High and low precision estimates were correlated for cued (*r*=0.6526; p=0.021) and uncued trials (*r*=0.5362; p=0.0589), whereby both relations exhibited a significant slope cued: a=1.8658 ($F_{(1,11)}$ = 15.867; p=0.0021) | uncued: a=1.9334 ($F_{(1,11)}$ = 10.737; p=0.0073). For this experiment, precision estimates did also not differ between cued and uncued trials (d=62.1927, p<0.001). The cued trials in experiment 3c (SOA40) showed a significant relation between high and low precision estimates as well (*r*=0.6353; p=0.0196) with a significant ratio of a=1.6376 ($F_{(1,12)}$ = 19.99; p<0.001). Likewise, the

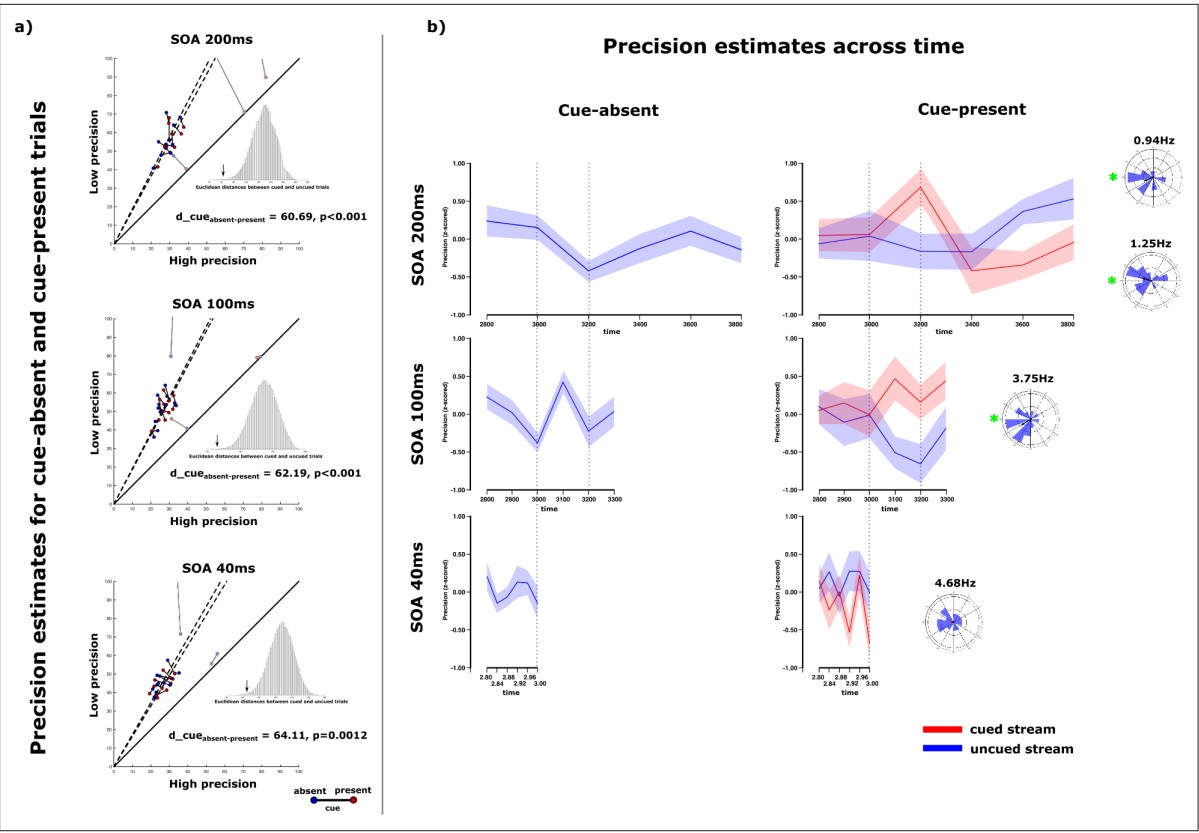

**Figure 4.** Results of Experiment 3. (**a**) The ratio of slightly lower than 2 remains present between high and low precision estimates for experiment 3. Importantly, individual precision estimates do not vary between trials in which a luminance cue is presented and in trials in which the luminance change is absent. The cue itself does not influence the general precision estimate collapsed over tracking intervals. This holds true for all three sets of tracking intervals (SOA200 /SOA100/SOA40). (**b**) Precision estimate time courses with standard errors (n = 15). For cue-absent trials no systematic variation in precision estimates could be observed over time in Exp3a (SOA200) and Exp3c (SOA100). However, precision estimates changed for different tracking intervals in cue-absent trials in Exp3b (SOA100). In Exp3a (SOA200), precision estimates for the cued and uncued stream differed over time. Specifically, precision for both streams seemed to exhibit an antiphasic relation at around 1 Hz. Phase differences of precision estimates between cued and uncued streams cluster around 180° in this case. Exp3b (SOA100) shows a very similar time-course for the precision estimates of cued and uncued streams over 2.8 sec to 3.2 sec comparable to the SOA200-experiment. Another, faster, oscillation (3.75Hz) can be additionally observed with the same anti-phasic relation between cued and uncued stream. In Exp3c (SOA40) no stream-related differences over the tested intervals can be observed.

estimates for the uncued trials were correlated ($r$=0.6069; p=0.0278) with a slope of a=1.7290 ($F(1,12)$ = 18.424 p=0.001). Both patterns of results were highly similar (d=64.1139, p=0.0012).

Collapsing across time throughout experiments 3 a-c, the pattern of precision estimates, including a ratio of slightly less than 2, confirms the finding of a general uneven distribution of feature-based resources so far. This ratio of consistent resource allocation does also seem to be uninfluenced by the presentation of a luminance cue shortly after onset of the feature change.

Next, precision estimates were calculated for each SOA increment (t1...t6) in trials without and in trials with a brightness cue (cue-absent, cue-present trials). For cue-present trials, precision estimates were obtained as a function of whether the cued or the uncued stream was tested. *Figure 4b* exhibits the precision time-courses for the cued and uncued stream for cue-present trials as well as the separate precision time-course for cue-absent trials in experiments 3 a-c.

## Experiment 3a

Precision estimates for cue-absent trials do not show any significant variation over the 200ms SOA increments ($F(5,70)$ = 1.676, p=0.153, e=0.107). For cue-present trials, the precision estimates over time every 200ms do not vary, as revealed by a non-significant main effect of SOA ($F(5,70)$ = 1.487, p=0.213, e=0.096). We find, however, a significant interaction between SOA and stream ($F(5,70)$ = 2.495, p=0.039, e=0.151) in the cue-present trials, while the main effect of stream remains not

significant (F(1,14) = 0.260, p=0.618, e=0.018). Hereby, precision estimates for cued and uncued streams differed at 3.2 sec (t(14) = –2.443, p=0.028, e=0.299) and 3.6 sec (t(14) = 3.001, p=0.010, e=0.391). The morphology of the precision time courses of cued and uncued color streams in cue-present trials suggest that the cue indeed induces a stream-specific phase reset, time-locked to the brightness increment. With time, the resource allocation then shifts systematically from the cued to the uncued stream. To further investigate the relation of resource allocation between the cued and uncued stream, both time-courses were Fourier-transformed, and the absolute phase-relation between the precision time courses of the cued and uncued stream was analyzed. At around 1Hz (0.94Hz and 1.25Hz) the phase difference between the precision time-courses for the cued and uncued stream showed a maximum phase-lock of 0.35 (0.94Hz) and 0.35 (1.25Hz). Crucially, the mean phase difference between the precision time-course of the cued and uncued stream is unimodally centered around 180° at 0.94Hz (v(14) = 4.95, p=0.035, m=199.87°) as well as at 1.25Hz (v(14) = 5.05, p=0.033, m=163.09°).

## Experiment 3b

The precision time-courses sampled at 100ms for cue-absent trials varies consistently over SOAs (F(5,70) = 2.87, p=0.021, e=0.17) (*Figure 4b*). For cue-present trials, precision estimates differ between the cued and uncued stream (F(1,14) = 11.095, p=0.005, e=0.442) but not over time (F(5,70) = 0.456, p=0.771, e=0.032). The interaction between stream and SOA is not significant (F(5,70) = 1.228, p=0.305, e=0.081), although precision is higher when testing the cued compared to the uncued stream from 3.1 sec onwards (3.1: t(14) = –3.103, p=0.008, e=0.407; 3.2: t(14) = –2.074, p=0.057, e=0.235; 3.3: t(14) = –2.133, p=0.051, e=0.245). Please acknowledge the overall similarity of the time-course of precision estimates of the cued and uncued stream between 2800 and 3200ms when sampling every 200ms or 100ms. Although experiment 3 a and 3b were performed by a different group of subjects, and covered a different temporal range, precision measures for both streams start to diverge consistently and substantially at about 3.1 sec after the brightness cue.

The Fourier-transformed precision time-courses exhibit a consistent phase difference between the cued and uncued stream at around 3.75Hz (PLV = 0.39), with the phase difference centering around 180° (v(14) = 5.12, p=0.031, m=211.51°) (*Figure 4b*).

## Experiment 3c

For SOA increments of 40ms no systematic variation in precision responses could be observed for the cue-absent trials over time (F(5,70) = 0.695, p=0.629, e=0.047). Cue-present trials as well did not exhibit any differences in precision as a function of SOA increment (F(5,70) = 1.242, p=0.299, e=0.081), stream (F(1,14) = 3.202, p=0.095, e=0.186), or interaction of both (F(5,70) = 1.345, p=0.256, e=0.088). The phase-lock for the phase differences between the Fourier-transformed time-courses of the cued and uncued stream peaked at around 4.68Hz (PLV = 0.18). The direction for the phase-differences at that frequency of 161.10° failed to be show a unimodal distribution around 180° (v(14) = 2.59, p=0.172) or 0° (v(14) = –2.59, p=0.828).

## Discussion

The reported experiments demonstrate that subjects are generally able to track two simultaneously and independently changing color streams over time. The results, however, show that at any moment, the precision of the representation of one stream is about twice as large as that of the other stream, suggesting that the attentional resources are unevenly distributed at a ratio of ~2:1. Critically, this ratio emerges consistently across different experimental conditions, different groups of subjects, and it does not depend on whether the color-values have to be reported sequentially in the same or separate trials. Finally, we find that the resource distribution between concurring color streams is not fixed but varies over the time at a speed of around 1 Hz, with resource allocation alternating between streams in anti-phase.

The alternation rate of ~1 Hz is substantially lower than oscillatory processes reported to underlie feature tracking in other work (*Re et al., 2019*). *Re et al., 2019* investigated the maintenance of two static color-values and argued for an oscillatory process at around 4 Hz with no representation-specific phase-shift. However, the internal feature-representations in this study did not traverse color space

such that color representations had not to be updated continuously. Here, we show that when such updating is necessary, simultaneous tracking of two color-streams involves a dynamic allocation of attentional resources that runs in antiphase. This process operates at much slower speeds, in that it takes around 500ms to turn the bias for one representation to the other. Hence, we assume that the performance fluctuation observed here is qualitatively different from the 4 Hz fluctuation reported in *Re et al., 2019*. The latter presumably reflects a modulation driven by a faster but non-specific attentional control mechanism, which has been proposed to arise from a general central sampler, that rhythmically distributes resources towards relevant objects in a top-down fashion (*VanRullen, 2016*; *Landau et al., 2015*; *Tomassini et al., 2017*). The here observed alternation of resource allocation in the 1 Hz range is also lower than attentional cycling processes documented in the spatial domain (*Landau and Fries, 2012*; *Fiebelkorn et al., 2013*; *Kienitz et al., 2018*; *VanRullen et al., 2007*). Here, performance fluctuations were found to cycle between spatially separated objects at a rate of 4 Hz (*Fiebelkorn et al., 2013*), which was proposed to reflect a periodic reweighting of attentional prioritization of those objects. Similar oscillatory sampling processes have been described using visual search paradigms varying in complexity (*Dugué and Vanrullen, 2014*). Ongoing perceptual and attentional mechanisms were shown to facilitate feature- and conjunction searches by sampling the relevant spatial information in a sequential manner at frequencies within the alpha and theta range, respectively (*Dugué et al., 2017*; *Dugué et al., 2016*). Attentional engagement is hereby indicated by a phase-reset within the relevant oscillatory mechanism (*Dugué et al., 2015*). Interestingly, very low-frequency oscillations around 2 Hz have been consistently reported. However, those frequencies represent the lower bound of spectral resolution in those studies (*Dugué and Vanrullen, 2014*; *Dugué et al., 2017*). Importantly, contrary to prior studies investigating oscillatory attentional engagements, the current work does not imply a temporal regularity (oscillation) to underlie the alternation between the two relevant color value representations.

In the present framework of feature-value-based tracking, a much slower rate of alternating between two feature representations is observed which, as outlined below, may reflect (1) the speed with which attention can reweight and sharpen feature representations and (2) the specific limitations of how attention is allocated in feature space. In both cases, the temporal structure of alternation would be variable and reflect the demands of individual feature-value selection.

## Feature attention is inherently slow

Using SSVEP recordings, it was found that a color cue biases color attention slowly towards one of two superimposed color RDKs (*Andersen and Müller, 2010*; *Forschack et al., 2017*), with strongest selectivity appearing 500–600ms after cue onset. Moreover, feature attention has been shown to rely on a sequence of spatiotemporal modulations in extrastriate visual cortex, unfolding as a coarse-to-fine tuning process over a period of 200ms – 400ms (*Bartsch et al., 2015*; *Bartsch et al., 2017*; *Bartsch et al., 2021*). Sharpened color selectivity (stronger tuning) is only attained late in the modulation phase. Hence, resolving the tracked colors with high precision, as required in the present experiments, entails sharpened tuning towards one or the other color, resulting in a comparably slow change rate. The temporal profile of this systematic, yet non-oscillatory, alternation of feature-based resource allocation would be modulated by the discriminability of the present color value within a non-linear color space.

## Feature-values can only be attended one at a time

A 2:1 ratio of tracking precision between streams alternating at ~1 Hz implies a strong resource limit, effectively allowing only one stream to be attended with precision at a given time. This interpretation dovetails with studies showing that subjects can only be consciously aware of – or access - one non-spatial feature value within a feature dimension at any given point in time (*Huang et al., 2007*; *Huang and Pashler, 2007*; *Huang, 2010a*; *Huang, 2010b*; *Houtkamp and Roelfsema, 2009*). A formal account of this limitation has been put forward by the Boolean map theory (*Huang and Pashler, 2007*), which posits that attentional selection works on a labelled Boolean map representation of the input. This sets specific limits on the way feature attention can operate. Specifically, the Boolean map is the representational format required for conscious access to feature-values, such that it makes the feature-value reportable. Such representation, however, can only be established

for one feature-value at a time. Multiple objects/locations defined by different feature-values can be represented, but then, the identity of the feature-values becomes inaccessible. Hence, the attempt to access multiple feature-values in parallel engenders a sequential process (feature-by-feature selection) (*Huang and Pashler, 2007*; *Morales and Pashler, 1999*; *Huang and Pashler, 2002*), that makes selection a comparably slow operation (*Huang et al., 2007*). *Morales and Pashler, 1999* demonstrate that symmetry judgments that are to be made based on the spatial distribution of color patches are accomplished by switching between colors. They find that RT increases from ~1200ms for two-color displays to ~2000ms for four-color displays, suggesting that the time to switch between colors takes something around 400ms, which is well in line with the alternation time seen here. The limitation of being able to access only one feature-value at a time, may account for the alternation in accessing the color streams when tracking over time. Research investigating object-based tracking of different features (*Blaser et al., 2000*) revealed that changing features of one objects can be easily tracked, whereas the simultaneous tracking of features of different objects results in poor performance. Importantly, analysis of the response pattern suggested that subjects did not divide but switched attention between objects when attempting to track the features of two objects simultaneously.

## Materials and methods

All experiments were approved by the ethics-commission of the Otto-von-Guericke University (no. 141/20). All participants gave written informed consent and consent to publish prior to their participation.

### Subjects

Eighteen subjects (15 female/ 3 male) with a mean age of 30.5 (SD = 7.09) participated in the first study. None of the subjects reported any psychological or neurological disorders and had normal or corrected-to-normal vision. All participants additionally confirmed correct color-perception. Subjects were monetarily compensated for their participation.

### General stimulus material

All stimulus material was generated using the Psychtoolbox for Matlab. Subjects were presented with a circular annulus with an outer radius of 5.66° consisting of two clouds of 900 colored dots each (0.2° in diameter), creating the perception of two superimposed objects occupying the same location (*Figure 1a*). The two objects' colors changed continuously throughout each trial.

The color space of the utilized monitor (Asus VG248QE) was calibrated using the measurements of a colorimeter (SpectroCAL, Cambridge Research Systems) from which the device-specific chromaticity space was calculated. The actual hues for the experiment were drawn from the normalized CIELUV space defined as a circle centered on the white point in a luminance plain of that space (u=0.1978, v=0.4683, 14.5 cd/m$^2$). The circle subtended a radius (i.e. chromaticity) of c=0.0576 for a standard sRGB gamut under D65 illumination. Thus, color values within this 1-dimensional circular space varied continuously in hue (from 0° to 360°) while being matched for luminance and chroma.

During each trial the colors of both dot objects moved continuously through the circular hue space independently of each other at a speed of 1°/frame. Those color trajectories were calculated offline in the following manner: Each individual trajectory started at an initial random angular position, subsequently moving with 1°/frame for 480 frames (8 sec) along a unit circle. At random intervals in between 30 and 100 frames (0.5 sec – 1.66 sec) the direction along the circle could revert, creating random, unpredictable but continuous movements within the circular hue space. Sets of several hundred trajectories were calculated and subsequently paired into two color streams for each trial. The only criterion for each pairing was the maintenance of a minimum distance between both trajectories of 60° at each frame to avert a perceptual confusion of both colors. This strategy was employed to avoid any direct interdependence between the histories of each of the stream pairs, since although colors would never cross (always larger than 60°), the actual angular distance at which one or the other stream would revert direction was entirely unpredictable.

At the initial generation of the dot clouds at the beginning of each trial, the order of presentation of individual overlapping dots was randomized in order to avoid creating a perception of depth with one object being located in front of the other.

Since cognitive processes maintaining solely feature-based information are being investigated, one goal was to control for any spatial information that could potentially be utilized by the subject during the feature-based tracking of the two streams. First, the annulus has an inner radius of 2.20°, restricting the primary task to peripheral vision only. Furthermore, a strategy of 'dot-flipping' was employed: At every frame throughout the continuous hue change, 10 random dots of both streams interchanged identity (*Figure 1a*), therefore removing any local spatial cues that might form during tracking, while leaving the color features (hue) and the amount of color information (900 dots each) intact.

## Stimulus analysis

In order to ensure that subjects were not able to predict or report object colors solely based on stimulus statistics, descriptions for the color streams within each trial across the entire length of 480frames (8 sec) for all 150 utilized trials were calculated. This includes the average amount of the combined hue-reversals amongst the two color-streams per trial as well as the overall distribution of hue distances between streams. Additionally, the amount of intersection between all the hue-values of the two streams should give an indicator of categorical separation of streams within hue-space for each trial. Next, the distributions of target colors (last presented hue before recall) for both streams were tested against a uniform distribution using Kolmogorov-Smirnov tests within each subject to investigate any potential bias for specific colors subjects have to report on. Note, that the distribution of actual target colors was specific for each subject, since although the same 150 pairs of color streams were used for all subjects, trial-length (the point at which hue-change seized and objects had to be reported on) varied randomly between 6–8 sec. Similarly, the distribution of hue-distances between the two target-hues of each trial were tested for uniformity for each subject using Kolmogorov-Smirnov tests.

On average 8.79 combined reversals (SD = 2.93) within the two color-streams were introduced randomly across the 8 sec timeframe. Such a reversal occurred on average after 0.7974 sec with a large variation of SD = 0.8228 sec, making them highly unpredictable. Across all trials and time points the angular distance between the two simultaneous color features was mostly evenly distributed between angles of 90° to 180° with the distances ranging between 60° and 90° being slightly less represented. This is to be expected with 60° being the lowest possible distance within hue-space the two color-streams could approach at any given time. Despite the introduction of a minimum possible inter-stream distance, absolute hue values between the streams were still fairly well shuffled across trial, in that a large portion of all hue-values of the two streams intersected for each trial (M=40.8%, SD = 32.8%).

The target hue-values (target colors at the time of report) were tested against a uniform distribution for each subject. All 300 targets were evenly distributed across the circular hue space in all but one subject (p=0.029), in which slightly more greenish (~180°) and purple (~240°) hues were present as targets. In all other 17 subjects, target hues were not different from a uniform distribution (0.335<p<0.993). Hue-distances between the two streams at the time of report exhibited an even distribution in between 90° and 180° degree in all (0.052<p<0.832) but two subjects (p=0.018, p=0.024), making statistical inferences for one of the target color based on the other highly unlikely.

## Experiment 1
### Procedure

Subjects were placed in a dark shielded chamber with a 24" FullHD LED monitor (Asus VG248QE) placed 70 cm in front of them. Throughout the experiment a fixation dot (0.12° diameter) was present and subjects were required to maintain fixation for most of the paradigm. At the start of each trial two superimposed colored cloud objects appeared on a black background and continuous hue changes in both objects were introduced as outlined above. Subjects were asked to attend to both color streams simultaneously as accurate as possible throughout the hue change. The trial continued for 6–8 sec, at which point both streams' colors changed to an isoluminant uniform grey ([72 72 72] RGB… white point of used color space…devoid of any chroma). With the removal of the hue information, a colored ring (color-wheel) appeared within the inner diameter of the annulus (outer radius: 1.36°, inner radius: 0.95°) representing the entire available 1D-circular hue space as well as a white cursor (0.07°) placed at a random position on the edge of that circle (*Figure 1b*). Now subjects had to perform a precision estimation task by reporting the last perceived color for both streams ($T_a$ and $T_b$)

as precisely as possible by moving the cursor along the colored ring using a manual dial (SpaceMouse Compact, 3Dconnexion). The cursor position was recorded as response as soon as the top of the dial was pressed ($R_1$). A second response was subsequently recorded the same way ($R_2$). As soon as two consecutive color estimation responses were entered, feedback was given for two seconds by highlighting two additional cursor positions (grey) at the two target colors ($T_a$ and $T_b$). After another 1 sec of blank screen the next trial started. Overall 150 trials were presented.

### Data analysis

The precision of the individuals' responses was operationalized as the standard deviation of the angular differences between target color and reported color (T-R). Assuming that continuously changing values within one feature dimension (color) are maintained by separate representations drawing on independent cognitive resources analogous to various fixed and variable slot or precision models in working memory (*van den Berg et al., 2012*; *Merkel et al., 2021*), the overall distribution of angular differences for each subject in experiment 1 should be a mixture of two separate precision distributions. Each one with a specific deviation, quantifying the amount of allocated feature-based resource for one of the two color streams, respectively.

During each trial, subjects had to perform two reports (one for each target color). Each of the two responses are potentially associated with one of the two color streams since both are equally relevant, resulting in two ambiguous target-response allocations (and therefore 2 pairs of precision responses) for each trial (($T_a$-$R_1$ & $T_b$-$R_2$) or ($T_b$-$R_1$ & $T_a$-$R_2$)). Therefore, assumptions have to be made about the subjects' intentions regarding the two given responses towards the two target colors. The post-hoc target-response assignments are non-trivial since both target values are equally relevant and their distance towards each other varies across trials. Additionally, response precision varies across trials and targets such that a higher resource allocation towards a specific target value can lead to a few poor responses due to natural fluctuations across trials. The pair of precision responses entered into the analysis was selected based on the minimum angular difference for the first response with any of the two targets (Min($T_x$-$R_1$) & ($T_{-x}$-$R_2$)) assuming that subjects performed the first report for one of the color targets usually with higher confidence and accuracy (*Figure 2a*). Thus, the first response was paired with whichever of the two targets was closest to that first response and the second response was paired with the remaining target. Note, that this assignment still allows for the accuracy (target-response distance) of the second response to be higher than the accuracy of the first response in any given trial. As a control, responses were paired with targets based on the overall minimum angular difference in one trial irrespective of the order of the given response (Min($T_x$-$R_y$) & ($T_{-x}$-$R_{-y}$)) yielding however the same pairings for most of the trials (94.78%+–3.03%).

The resulting response distribution for each subject containing 300 angular response differences (2 per trial) ranging from –180° to +180° was fitted with a mixture model of two von Mises distributions using an expectation maximization algorithm. Parameter estimation was restricted for each von Mises distribution being centered on 0°(+–10° to compensate for angular biases generated by random noise). Furthermore, each of the von Mises distribution was restricted to explain exactly 50% of the entire model, since 150 of the 300 angular differences are related to the responses towards one of the two color targets. The actual precision of the responses and therefore the precision of tracking for the two color-streams were modeled by the estimate of the standard deviations (sqrt(1/kappa)) for the two von Mises distributions (*Figure 2b*).

In order to validate the two precision estimates for each subject, Monte-Carlo simulations based on 1000 response distributions were performed. The motivation was to confirm that performing that particular tracking task with a specific pair of precision values (or ground truth) causes our mixture-model to estimate that exact pair of precision values. Two sets of 150 target-response differences were calculated by taking the target angles of a particular subjects and adding response angles based on random draws of precision responses from two von Mises distribution with the previously estimated standard deviations for that subject. Thus, each of the 1000 simulations per subject yields a pair of precision estimates, for a theoretical observer who responds to each pair of targets with two responses drawn from independent precision distributions (ground truth) with the observed standard deviations.

The simulation thus creates a two-dimensional null-distribution for any given pair of precision parameters. This null-distribution was modeled for each subject using a two-dimensional (4-parameter)

Gaussian (*Figure 2c*). The position of the observed pair of precision parameters within the null-distribution determined therefore the deviation from the proposed model (significance p).

For each subject the pair of precision estimates could be ordered into a low and high precision value. Those were subsequently correlated to test whether there is an underlying relation between the amounts of resources allocated to multiple feature-based representations across subjects.

## Experiment 2

### Stimulus and procedure

Sixteen of the subjects participating in the first experiment took part in experiment 2 (14 female / 2 male, mean age 30.69 (SD = 7.53)). The current control experiment did not differ from the first paradigm except for one crucial point: At the end of each hue-changing phase of each trial, only one of the two streams changed its color to an isoluminant grey [72, 72, 72] while the other stream maintained the last hue of the color stream (*Figure 1b* – Exp2). At that point a color wheel was presented, as in the previous study, and subjects were required to move a cursor along that wheel using a manual dial to indicate the last perceived color of the now grey stream. Following a confirming press of the dial, subjects received feedback through a second cursor appearing in grey at the actual target-color location on the color wheel.

The same set of trajectories was used as in the previous study to generate the hue-changes. However, the length of each trial was again drawn from a uniform random distribution between 6 and 8 sec. A total of 150 trials were presented and for each of them an unambiguous target-response difference for the report of the last perceived color of the grey target-stream was recorded.

### Data analysis

The distribution of all 150 angular target-response differences was modeled to be derived from the representational precision of two separate feature-based cognitive resources. Hereby during each trial exactly one of the two resources (allocated to the two color-streams) are probed. The association of the reported color stream and one of the two resources cannot be known and is random. Therefore, across the experiment it is to be expected that around 50% of each of the two resources is going to be a target. A mixture model of two von Mises distributions was used to estimate the precision of maintaining each of the two feature representations and their allocated resources over time. Models were restricted in that each had to explain 50% of the variance of the data. Furthermore, both von Mises distributions were centered on 0°(+−10°). Cognitive resources allocated to each of the two feature representations were quantified as the estimated standard deviations (sqrt(1/kappa)) of the two fitted von Mises distributions.

Those precision estimates were again validated using Monte-Carlo simulations with the same logic as in the first experiment. For those simulations, 1000 distributions of 150 target-response differences were created from the sum of the actual target-colors for the particular subject and 75 random draws from two different von Mises distributions with the previously estimated standard deviations for that subject. Pairs of precision estimates from fitting the mixture model to each of the simulated target-response distributions constituted 2-dimensional null-distributions of precision values for each subject and, as in the previous analysis, determine the significance of the observed precision values for each subject. The relation between high and low precision values across subject was quantified using linear regression.

## Experiment 3

### Subjects

Forty-five subjects, who did not take part in any of the previous studies and were naïve to the task, participated in the current set of experiments. For three iterations of the experiment in which different SOAs between color change onset and response were introduced, fifteen different subjects each participated. For the SOA-200ms version (Exp3a), mean age of the subjects was 25.07 (SD = 4.83) with eleven females and four males participating. Exp3b in which duration between color onset and response varied in steps of 100ms, nine females and 6 males participated with mean age of all subjects being 23.60 (SD = 2.64). For the 40 ms version (Exp3c), another nine females and six males with a mean age of 23.60 (SD = 2.77) participated.

## Stimulus and procedure

An annulus stimulus consisting of two overlapping set of dots was used with the same visual properties as in the previous set of experiments. Each trial started out with the hue of both streams being uniform grey ([72 72 72]) for 1–1.5 sec. Next, the two sets of dots appeared in different hues, which subsequently moved through color space. New pairs of trajectories along the hue circle were calculated offline with the same constraints as in the previous experiments. After the period of hue change, one of the targets turned grey again, at which point the color-wheel within the annulus appeared and subjects had to report the last perceived hue of the probed stream as precisely as possible using a dial.

Crucially, in the current design, the duration of the motion through hue space until one of the target streams was probed (turned grey) was introduced as an additional factor and varied along six SOAs (*Figure 1b* – Exp3). This way, the resource distribution amongst the two streams (as quantified by the precision estimates for the last perceived hue of the probed stream) can be analyzed as a function of a systematic SOA-variation between the color-change onset and response. As potentially relevant time-varying processes could be located at different frequencies within the spectral domain, three separate paradigms were conducted with 15 subjects each. Hereby, the factor 'time' had different sets of levels for each iteration of the experiment, potentially covering different spectra of precision-variation. In the SOA-40-version of the experiment, the target-hue was probed at 3.00,3.04,3.08,3.12,3.16 and 3.20 sec after hue-onset. The theoretical spectral limit for resolving any meaningful process can hereby by considered 5Hz-12.5Hz, given the Nyquist-frequency determined by the sampling (40ms) as well as the overall time-range (200ms). For the SOA-100-version, change-durations were spaced at 100ms intervals starting from 3 sec (3.0,3.1,3.2,3.3,3.4,3.5) with a frequency limit of 2Hz-5Hz. Finally in the SOA-200-version, the 'time'-levels spanned 1 sec from 3 to 4 sec (3.0,3.2,3.4,3.6,3.8,4.0) with a spectral limit of 1Hz-2.5Hz. Not one subject performed in different iterations of the same task.

Additionally, in half the trials a brief luminance change (cued trials) was introduced (+15 RGB) in one of the two color streams 200ms after hue-onset for 67ms. In the other half of trials that were uncued, no such luminance change occurred in any of the color streams. Within the cued trials, either the stream in which the cue appeared was probed at the end of the trial (cued stream) or the stream that was not cued (uncued stream) was probed at the end of the trial. Both conditions in the cued trials were counterbalanced. Thus, the luminance cue itself did not carry any information about which stream was to be probed. The intention of the cue was to shift feature-based attention towards one of the two streams and introduce an object-specific phase-reset of the process involved in maintaining feature representations time-locked to the cue. Precision can now be recorded for reporting either the cued or uncued stream within the cued trials. Importantly, the luminance cue, while potentially modulating the temporal structure of the precision responses, is not suspected to change the average precision responses across the entire tested time range.

In order to gather enough responses to estimate reliable precision values for cued and uncued streams across all six timepoints, the experiment was partitioned for each subject into three separate sessions on 3 successive days. Across sessions, overall 36 trials per timepoint and stream (cued stream / uncued stream) in the cued trials and 72 trials per timepoint in the uncued trials were presented for a total of 864 trials. Subjects' task was to keep track of the two color-streams as precisely as possible and report the color of the probed (grey) stream at the end of each trial using a dial. They were told to not focus on any other aspect of the task. The appearance or significance of the luminance change was specifically not addressed by the experimenter throughout the sessions, but subjects were encouraged to simply ignore anything that did not concern their primary task if they mentioned it.

## Analysis

First, using von Mises mixture models, two pairs of precision estimates were calculated for each subject across SOAs and streams for the cued and uncued trials separately. For both trial-types (cued/uncued), the distribution of 432 target-response angular differences was fitted with the mixture model of two von-Mises distributions. The resulting deviation estimates quantified the low and high resource precisions for the trials in which one stream was cued and trials in which none of the streams were cued. This first analysis was used to replicate the previous finding of a general ratio of resource distribution amongst two relevant features within one dimension. In order to test, whether this ratio of resource allocation was altered by the appearance of the luminance-cue, the consistency of precision

pairs for cued and uncued trials within each subject was determined using the sum of Euclidean distances between precision estimates for cued and uncued trials across subjects (see Exp2). This analysis was performed for each iteration of the experiment separately (SOA40, SOA100, SOA200). Linear regression models without intercept were used to quantify the ratio between high and low precision pairs for cued and uncued trials.

Next, the time-course of response precision for the cued and uncued trials for each subject was estimated by fitting a half-normal distribution to the absolute target-response differences at each timepoint of the uncued trials and at each timepoint for the cued stream and uncued stream of the cued trials. The half-normal function was used to disregard negligible counter-/clockwise hue response biases and improve the fit. Note, that no mixture-model was used in this analysis, as the cue allows for an identification and therefore unambiguous separation of the streams. The resulting standard deviations were z-normalized to account for large variations in general performance across subjects. Using multi-factorial rANOVAs, variations in response precision across trials(cued/uncued), streams(cued/uncued) and crucially SOA(t1...t6) were analyzed for the different iterations of the experiment (SOA40/SOA100/SOA200). rANOVAs were adjusted for multiple comparisons using the Greehouse-Geisser correction.

Finally, precision time-courses (along SOAs) for the uncued trials as well as the cued and uncued streams of the cued trials were Fourier-transformed. To analyze systematic precision variations over time for the two continuously maintained color streams, phase-differences were calculated between cued and uncued streams of the cued trials and tested for non-uniformity against 0° (in-phase) and 180° (counter-phase) using v-tests at frequencies exhibiting peak phase-lock.

## Acknowledgements

The current study was supported by the Deutsche Forschungsgemeinschaft grant SFB1436/B05.

## Additional information

### Funding

| Funder | Grant reference number | Author |
|---|---|---|
| Deutsche Forschungsgemeinschaft | SFB1436/B05 | Jens-Max Hopf |

The funders had no role in study design, data collection and interpretation, or the decision to submit the work for publication.

### Author contributions
Christian Merkel, Conceptualization, Data curation, Software, Formal analysis, Validation, Investigation, Visualization, Methodology, Writing - original draft, Writing - review and editing; Luise Burgmann, Mandy Viktoria Bartsch, Data curation, Formal analysis, Investigation; Mircea Ariel Schoenfeld, Supervision, Writing - original draft, Project administration, Writing - review and editing; Jens-Max Hopf, Supervision, Funding acquisition, Writing - original draft, Project administration, Writing - review and editing

### Author ORCIDs
Christian Merkel ⓘ http://orcid.org/0000-0002-8730-5653
Mandy Viktoria Bartsch ⓘ http://orcid.org/0000-0002-9276-5160

### Ethics
All experiments were approved by the ethics-commission of the Otto-von-Guericke University (no. 141/20). All participants gave written informed consent and consent to publish prior to their participation.

### Decision letter and Author response
Decision letter https://doi.org/10.7554/eLife.91183.sa1
Author response https://doi.org/10.7554/eLife.91183.sa2

## Additional files

### Supplementary files
• MDAR checklist

### Data availability
All raw data the analyses are based on as well as analysis scripts are available online via the OSF repository.

The following dataset was generated:

| Author(s) | Year | Dataset title | Dataset URL | Database and Identifier |
|---|---|---|---|---|
| Merkel C | 2023 | Multiple feature-value tracking | https://osf.io/y3qst/ | Open Science Framework, y3qst |

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
