## [Editor Report]

This study presents an important finding on how human observers keep track of continuously changing feature values across two different streams but within the same dimension. The conclusion about the serial attentional resource allocation during parallel feature value tracking is informative to understand the function of visual cortical systems. The experimental evidence supporting the conclusion is convincing.

---

## [Decision Letter]

**Decision letter after peer review:**

Thank you for submitting your article "Serial attentional resource allocation during parallel feature value tracking" for consideration by *eLife*. Your article has been reviewed by 3 peer reviewers, including Xiling Zhang as Reviewing Editor and Reviewer #1, and the evaluation has been overseen by Joshua Gold as the Senior Editor.

*Reviewer #1 (Recommendations for the authors):*

I suggest that the authors first explain the null difference between cued and un-cued conditions in Exp. 3s, and clarify how unbiased and adequate assign response 1 and response 2 to target 1 and target 2, respectively.

*Reviewer #2 (Recommendations for the authors):*

1. I wonder if the classification of two possible target-response distances in Experiment 1 is a valid approach. The manipulation in Experiment 2 by randomly testing one of the two color streams seems better, but it is not clear how to separate the high-precision and low-precision distributions can be separated from a mixture distribution.

2. A ratio of 2:1 seems to be quite robust across different experiment setups. We need to be careful to see if this does not reflect a bias due to computational processes. Perhaps some simulations would help confirm it.

3. In Experiment 3a-c, the authors performed several versions of the experiment with different SOAs. The consistent phase difference between the cued and un-cued streams also varied with SOA (~1Hz for 200-ms SOA, 3.75 Hz for 1o0-ms SOA; 4.68 Hz for 40-ms SOA). It is not clear to me why the authors only made a conclusion based on 1Hz oscillation between two streams.

4. The authors should discuss more about the functional significance underlying the observation of "unequally distributed attentional resources that alternated at 1 Hz" with two attended feature values.

*Reviewer #3 (Recommendations for the authors):*

Below I am describing my concerns in more detail and making suggestions of how to address some of them.

1) The task instructs participants to "keep track" of two colored dot clouds and asks them to report the color after about 6-8 seconds on each trial. Certainly, there is some uncertainty with regards to when color is probed, yet it seems to me that participants can easily enough start attending to the color shortly before the test probe at the end (so, after about 6s), to be able to report the colors. Thus, the task itself does not truly require participants to track color values continuously for several seconds. The task can be performed by starting to pay attention to the colors only at the very end, shortly before the test probe (at least in Exp. 1 and 2). Also, as participants choose which color to report, it is not necessary for them to know which stream had which color – so "swaps" might be happening all the time; based on Exp. 1 it's entirely unclear which color stream they are intending to report.

Overall, this task design is very different from spatial attention tracking tasks that require participants to track target objects among identical distractors. The color tracking task in the current study tries to establish parallels between the spatial tracking literature, but it seems not clear at all where the parallels are given the differences in tasks. Spatial tracking tasks (multiple-object tracking, i.e.,) require selective and continuous tracking (e.g., Stormer et al., 2014), which is not true on the current color task that asks participants to report which colors they just saw.

Thus, it is unclear whether or to what extent this task tests "tracking" vs. simply perceiving and briefly remembering two colors once the test probe comes on, more akin to a visual working memory task with a short delay.

One potential additional analysis that could – at least to some extent – get at this might be to examine the response errors in relation to not just the very last color, but previous colors of that stream and examine recency effects/biases in people's color reports. If people are continuously tracking each color stream you'd expect relatively strong recency biases, consistent with "serial dependence" (Fischer and Whitney, 2014). Another paper on visual working memory for continuously color changing objects also seems relevant: Chung, Schurgin, Brady, 2023.

2) This relates to my previous point that questions the task itself but relates more directly to the main claim of a limited resource. Given participants are asked to report the color at the very end of the trial, this measure – in my view – might well reflect memory and decision-related processes, rather than attentional allocation. How can we tease these processes apart in the current task design? In some ways, it seems to me, that the probe at the end of the trial forces people to prioritize one color over the other, so the limits the authors are claiming might all happen at the response stage and not during tracking/viewing.

3) I am not sure I follow the logic of the correlation analysis and interpretation; what they seem to show is that the first and second responses are correlated, but I am not sure how we can conclude from this pattern that there is competition between attending to the two streams. Such a model would predict that a response on one trial trades off with the response on that same trial in particular, no? So, shouldn't these trade-offs be measured at a single trial level, rather than accuracy across people for two responses? It seems to me these correlations are consistent with a much simpler explanation that considers overall differences in performance between people (e.g., those people that have the best high-precision responses also have the best low-precision responses – thus, are overall more precise than other people), and that the second response is always a little worse than the first (probably due to longer memory delay). Please clarify the logic behind the analyses.

4) Experiment 3 used salient cues to one stream to direct participants' attention to that stream, and then asked participants to report the colors at different cue-probe intervals. I appreciated the higher sampling and larger unpredictability here f which likely introduced more of a feature tracking task than Exp. 1 and 2.

The authors find some differences in performance across the different SOAs, and in particular in Exp. 3A, find an interaction between cue condition and SOA tested, revealing higher performance at about 3.2s after the cue for the attended stream which then switches back to higher performance for the uncued stream. None of the other two experiments find interactions that would support that only one stream is monitored at a time. Given the authors had no a priori expectation of when such switches happen I would really like to see a replication of Exp. 3A to ensure that the one effect they observed here is not a false positive. Furthermore, while some temporal structure might be present in this data (if it replicates), making the additional step and concluding that this is an 'oscillation' is a far stretch. All I see in the data is that there is an increase in performance at 3.2s for cued which then disappears again. This is not evidence for an oscillation. Brookshire, in a recent paper published in Nature Human Behavior, discusses the problem of applying spectral analyses to data such as the one here, as these analyses are not only sensitive to periodic but also aperiodic structure. There are several additional analyses that can be done to test this (see Brookshire paper for more details).

5) All data is analyzed using a mixture model that is often used in visual working memory tasks that relies on strong assumptions, namely that response errors reflect a mixture of variably precise memories and guesses. While the mixture model has been quite popular in working memory research, recent studies challenge the model's assumptions and support a much more simple framework in which all response errors can be explained with a single parameter, memory strength, once the psychological similarity of the feature space is considered (e.g., Schurgin et al., 2020). Thus, I would strongly encourage the authors to report and analyze the mean error and standard deviation of the errors as indexes of performance in this task instead of modeling the data.

6) The discussion of how the results relate to the Boolean map theory was interesting and I appreciated how the authors linked their current results to that theory. I was wondering whether the authors are arguing that the bottleneck assessed in the current task is related to "access" of the colors, i.e., the ability to report the colors, or a limit of attentional allocation to early sensory representations.

References:

Brookshire, G. (2022). Putative rhythms in attentional switching can be explained by aperiodic temporal structure. Nature Human Behaviour, 6(9), 1280-1291.

Chung, Y.H., Schurgin, M.W. and Brady, T.F. The role of motion in visual working memory for dynamic stimuli: More lagged but more precise representations of moving objects. Atten Percept Psychophys 85, 1387-1397 (2023).

Fischer, J., and Whitney, D. (2014). Serial dependence in visual perception. Nature neuroscience, 17(5), 738-743.

Schurgin, M. W., Wixted, J. T., and Brady, T. F. (2020). Psychophysical scaling reveals a unified theory of visual memory strength. Nature human behaviour, 4(11), 1156-1172.

Störmer, V. S., Winther, G. N., Li, S. C., and Andersen, S. K. (2013). Sustained multifocal attentional enhancement of stimulus processing in early visual areas predicts tracking performance. Journal of Neuroscience, 33(12), 5346-5351.

---

## [Author Response]

Reviewer #1 (Recommendations for the authors):I suggest that the authors first explain the null difference between cued and un-cued conditions in Exp. 3s

We further clarified in the methods the significance of the “cue” as a luminance change in order to phase-reset the parallel feature-tracking in contrast to a stimulus designed to continuously enhance responses towards that particular color stream in which it occurs.

Clarify how unbiased and adequate assign response 1 and response 2 to target 1 and target 2, respectively.

We ran an additional simulation (shown under reviewer 3) to showcase the outcome of the two potential target-response pairing procedures in experiment 1, outlined above. These calculations suggest an overestimation of the precision ratio between the two streams using the best/worst assignment strategy. Additional clarifications have been added to the methods.

Reviewer #2 (Recommendations for the authors):1. I wonder if the classification of two possible target-response distances in Experiment 1 is a valid approach. The manipulation in Experiment 2 by randomly testing one of the two color streams seems better, but it is not clear how to separate the high-precision and low-precision distributions can be separated from a mixture distribution.

The reviewer raises the issues of response interference when testing both streams. In fact, the second experiment serves as a validation of the separation-strategy while avoiding such interference problems. To further clarify this, we provide an extended description of our reasoning behind target-response assignments for experiment 1 in the methods section. Additionally, the logic has been laid out in more detail for reviewer 1 as well as below for reviewer 3. Importantly, the ratio measured by both methods (report both streams/ report one random streams) are comparable.

Furthermore, the reviewer raises an important point about the validity of mixture-modeling using von-Mises distributions. Separating a single distribution of precision-responses (target-response differences) into two von-Mises distributions based on the assumption of two underlying continuous resources contributing equally to the sum of all precision-responses is a well-established technique, employed extensively in the working-memory literature (Zhang, 2008; Bays, 2009; Fougnie and Alvarez, 2011, van den Berg, 2012). In our case, precision quantifies the successful access to information of one of two independent color perceptions potentially maintained by feature-based attention.

2. A ratio of 2:1 seems to be quite robust across different experiment setups. We need to be careful to see if this does not reflect a bias due to computational processes. Perhaps some simulations would help confirm it.

We thank the reviewer for motivating an analysis about the computational validity of the performed mixture model. We ran a simulation for experiment 2 with the null-hypothesis of both streams receiving the same amount of cognitive resource (being equally accessible) at any point in time. Using our employed Monte-Carlo approach described in the paper we created a number of theoretical target-response distributions consisting of mixture responses originating from von-Mises distributions with equal standard deviation. Over 1000 simulations were run with deviations based on our measured precisions for each subject. For each of those subjects we assumed each stream to be maintained by a resource quantified by the mean of the two prior estimated deviations ([55.27,25.70]->40.49, etc.). Basically, we tested how the model would behave if the real precision data in figure 3 would be positioned on the diagonal x=y, i.e. ratio = 1.

Separating those equal distributions using our mixture model resulted in a range of ratios presented in the following figure. The maximum number of results equal a ratio of 1 and the median reaches a ratio of about 1.24, with less than 5% of the simulated results reaching a ratio of more than 1.928. We are therefore confident, that the data in our results show a real imbalance in cognitive resources across both streams throughout each trial. This analysis has been added to the manuscript.

3. In Experiment 3a-c, the authors performed several versions of the experiment with different SOAs. The consistent phase difference between the cued and un-cued streams also varied with SOA (~1Hz for 200-ms SOA, 3.75 Hz for 1o0-ms SOA; 4.68 Hz for 40-ms SOA). It is not clear to me why the authors only made a conclusion based on 1Hz oscillation between two streams.

The reviewer is correct, that we did focus on the 1Hz modulation of the phase difference in experiment 3. Experiment 3 was designed in three different SOA versions in order to be able to replicate significant phase differences across subject groups. This replication was successful for the 1Hz modulation only. Hence, we are confident of that specific finding. For 3b this effect emerges as a significant precision difference between the cued and uncued stream from 3.1sec onwards. In this case, the range of the SOAs do only cover one half of a full period. The 3.75Hz phase-lock found in experiment 3b however did not replicate in the 40ms version of the task.

Furthermore, the maximum phase-lock at 4.68Hz in the 40ms-version has been reported for the sake of detailed reporting of the results but is rather spurious, showing minimal phase-lock of 0.18 and no significant in-phase- or antiphase- behavior.

4. The authors should discuss more about the functional significance underlying the observation of "unequally distributed attentional resources that alternated at 1 Hz" with two attended feature values.

We expand in the discussion more on the potential attentional process alternating between to color representations within a certain time-frame. Additionally, we discuss the importance of the methodological and cognitive dissociation of oscillations in response to reviewer 3.

Reviewer #3 (Recommendations for the authors):Below I am describing my concerns in more detail and making suggestions of how to address some of them.1) The task instructs participants to "keep track" of two colored dot clouds and asks them to report the color after about 6-8 seconds on each trial. Certainly, there is some uncertainty with regards to when color is probed, yet it seems to me that participants can easily enough start attending to the color shortly before the test probe at the end (so, after about 6s), to be able to report the colors. Thus, the task itself does not truly require participants to track color values continuously for several seconds. The task can be performed by starting to pay attention to the colors only at the very end, shortly before the test probe (at least in Exp. 1 and 2). Also, as participants choose which color to report, it is not necessary for them to know which stream had which color – so "swaps" might be happening all the time; based on Exp. 1 it's entirely unclear which color stream they are intending to report.Overall, this task design is very different from spatial attention tracking tasks that require participants to track target objects among identical distractors. The color tracking task in the current study tries to establish parallels between the spatial tracking literature, but it seems not clear at all where the parallels are given the differences in tasks. Spatial tracking tasks (multiple-object tracking, i.e.,) require selective and continuous tracking (e.g., Stormer et al., 2014), which is not true on the current color task that asks participants to report which colors they just saw.Thus, it is unclear whether or to what extent this task tests "tracking" vs. simply perceiving and briefly remembering two colors once the test probe comes on, more akin to a visual working memory task with a short delay.

We clarified details of the experimental design of experiment 1 and 2 in the methods section. Importantly, subjects either had to report *both* streams (exp1) or *one* random stream (exp2). In both experiments, in order to perform well, subjects had to try to maintain both color information at any given time as accurately as possible. None of the tasks let them ‘choose’, which color to respond to. As a similar point is mentioned by the reviewer later on again: In the first two experiments, we presume that one attentional resource is split between color streams at the start of each trial to maintain its changing information. Importantly, we cannot know which stream receives which resource at the start of the trial (this is where the logic of the ‘reset’-cue in Exp3 comes into play). Thus, the naming convention of ‘stream1’ and ‘stream2’ is arbitrary and only useful to make clear that two colors had to be attended to simultaneously. As the reviewer states, we cannot know which report corresponds to which target color (stream) in experiment 1. This is why we apply the described assignment-strategy that presumes that the first response is performed more confidently than the second in most cases. Since there is still a large amount of variability in this response strategy per trial we collapse all responses and separate them using a mixture model. We use the Monte-Carlo simulation in order to show that the estimated target-response precision pairing can be recreated using the assumption of responding to the ‘better’ maintained color stream first.

As an additional analysis, we also paired target and responses in a way to separate ‘best’ and ‘worst’ responses independent of the sequence of response. Interestingly, the Monte-Carlo simulation with this assumption did systematically overestimate the precision ratio relative to the initial precision pairs for each subject (see Author response image 1: left side depicts the analysis as seen in figure 3a of the manuscript; right side shows precisions estimates for exp1 under separation based on best/worst strategy (blue dots depict precision estimates and ellipses the confidences)). This means that only analyzing best and worst precisions would often pair a response with a target, that would not be the intended target but just randomly be closer to that particular target in this trial.

**Author response image 1. sa2fig1:** 

For example, let R1 pair with T1 and R2 with T2 at a given trial, with T1 = 0° and T2 = 90°. Let R1 be fairly accurate with R1 = 30° (diff = 30°) and R2 inaccurate with R2 = 20° (diff = 70°). Due to the random target configurations however, the ‘best’/‘worst’ strategy would automatically pair R2 with T1 (diff = 20°) and R1 with T2 (diff = 60°), leading to an overestimation of the ratio. Importantly, the second and third experiment were designed with one response required per trial as to avoid the mentioned problem of target-response assignment.

One potential additional analysis that could – at least to some extent – get at this might be to examine the response errors in relation to not just the very last color, but previous colors of that stream and examine recency effects/biases in people's color reports. If people are continuously tracking each color stream you'd expect relatively strong recency biases, consistent with "serial dependence" (Fischer and Whitney, 2014). Another paper on visual working memory for continuously color changing objects also seems relevant: Chung, Schurgin, Brady, 2023.

This is a very intriguing idea proposed by the reviewer. Indeed, we could assume, that the color representation we have access to at any given point in time ‘lacks’ behind the continuous color stream. Especially, since we propose a fairly slow model of alternating between the two streams. We therefore could look at precision as a function of time.

The problem with the current dataset, however, is that the sequence of color-trajectory presentations across the experiment for each subject was randomized, such that we can not recover the specific history for the color streams for any given precision response but only the very last target color. If we would know whether the color ‘came from’ clockwise or counterclockwise, we could look for a bias of the precision distribution indicating a temporal lack of the representation reported. However, this is a very intriguing idea that we will keep in mind for future studies.

2) This relates to my previous point that questions the task itself but relates more directly to the main claim of a limited resource. Given participants are asked to report the color at the very end of the trial, this measure – in my view – might well reflect memory and decision-related processes, rather than attentional allocation. How can we tease these processes apart in the current task design? In some ways, it seems to me, that the probe at the end of the trial forces people to prioritize one color over the other, so the limits the authors are claiming might all happen at the response stage and not during tracking/viewing.

The reviewer is right in that every precision task is a mixture of variations in attentional processing but also decision related processes. In the current task however, there should not be any systematic decision-related variation between responding to one stream vs responding to the other. Especially for the 2^nd^ and 3^rd^ experiment, in which subjects do not know, which of the two streams is being probed and therefore no prioritization does occur. Also, one has to see that an eventual prioritizing effect of probing one stream is orthogonal to the distributional variation of precision within the probed stream. In other words, if color prioritization due to probing at the end of a trial accounts for the performance variation, there should be no 2:1 precision variation among the probed colors, as they would all be equally prioritized above the unprobed color stream.

The design for the 2^nd^ and 3^rd^ experiment was chosen apart from avoiding the problem of target-response assignments, in order to exclude variance due to memory-related decay of the second response.

3) I am not sure I follow the logic of the correlation analysis and interpretation; what they seem to show is that the first and second responses are correlated, but I am not sure how we can conclude from this pattern that there is competition between attending to the two streams. Such a model would predict that a response on one trial trades off with the response on that same trial in particular, no? So, shouldn't these trade-offs be measured at a single trial level, rather than accuracy across people for two responses? It seems to me these correlations are consistent with a much simpler explanation that considers overall differences in performance between people (e.g., those people that have the best high-precision responses also have the best low-precision responses – thus, are overall more precise than other people), and that the second response is always a little worse than the first (probably due to longer memory delay). Please clarify the logic behind the analyses.

The reviewer’s interpretation is exactly correct. All we wanted to show with the correlation analysis is, that the ratio between the estimated precisions for both resources are comparable across subjects. This “trade-off” is hereby independent of the overall precision of responses. Subjects can be very accurate in responding to the targets but would still show a bias by responding better to one of the streams compared to the other. Importantly, this bias is not constant (as with the suggested memory delay which would be relevant only in the first experiment) as would be indicated by a similar precision offset, but by a constant ratio (1.7-1.9).

4) Experiment 3 used salient cues to one stream to direct participants' attention to that stream, and then asked participants to report the colors at different cue-probe intervals. I appreciated the higher sampling and larger unpredictability here f which likely introduced more of a feature tracking task than Exp. 1 and 2.The authors find some differences in performance across the different SOAs, and in particular in Exp. 3A, find an interaction between cue condition and SOA tested, revealing higher performance at about 3.2s after the cue for the attended stream which then switches back to higher performance for the uncued stream. None of the other two experiments find interactions that would support that only one stream is monitored at a time. Given the authors had no a priori expectation of when such switches happen I would really like to see a replication of Exp. 3A to ensure that the one effect they observed here is not a false positive. Furthermore, while some temporal structure might be present in this data (if it replicates), making the additional step and concluding that this is an 'oscillation' is a far stretch. All I see in the data is that there is an increase in performance at 3.2s for cued which then disappears again. This is not evidence for an oscillation. Brookshire, in a recent paper published in Nature Human Behavior, discusses the problem of applying spectral analyses to data such as the one here, as these analyses are not only sensitive to periodic but also aperiodic structure. There are several additional analyses that can be done to test this (see Brookshire paper for more details).

Please see the public section of the revision regarding the temporal structure of our proposed mechanism. We do absolutely agree, and emphasize in our paper, that attention in this case is not an oscillatory process but an alternation of conscious access towards one of two changing feature values over time as proposed by the Boolean map theory. We use Fourier-analysis based testing in order to merely quantify the time-frame in which these alternations occur, but not to imply oscillations and regularity. Two variations of experiment 3 do show time-based variations over a similar time-span with 3a over one period across 1 second and 3b over half a period across 500ms.

5) All data is analyzed using a mixture model that is often used in visual working memory tasks that relies on strong assumptions, namely that response errors reflect a mixture of variably precise memories and guesses. While the mixture model has been quite popular in working memory research, recent studies challenge the model's assumptions and support a much more simple framework in which all response errors can be explained with a single parameter, memory strength, once the psychological similarity of the feature space is considered (e.g., Schurgin et al., 2020). Thus, I would strongly encourage the authors to report and analyze the mean error and standard deviation of the errors as indexes of performance in this task instead of modeling the data.

We absolutely agree on the mentioned limitations of the mixture model in terms of variance separation into memory performance and guesses. However, here we do not make strong assumptions about the composition of variation explained by the model. We take ‘precision’ as a general reflection of a continuous but variable attentional resource. We do not go into interpretations regarding sources of ‘error’ but are content with the notion that a variety of errors contribute to larger or smaller precisions over a large amount of trials.

Experiments 2 and 3 require subjects to perform exactly one response per trial from which two precision distributions are estimated from. Therefore, non-modeled standard deviations would only give you the overall precision responses over all trials and all color streams and not be useful in distinguishing separate attentional resources for two changing color values. In experiments 3a-c we do report standard deviations without using a mixture model since we do have an unambiguous target-response assignment for cued and uncued streams.

6) The discussion of how the results relate to the Boolean map theory was interesting and I appreciated how the authors linked their current results to that theory. I was wondering whether the authors are arguing that the bottleneck assessed in the current task is related to "access" of the colors, i.e., the ability to report the colors, or a limit of attentional allocation to early sensory representations.

This is a very interesting question discussed extensively by Huang and Pashler. The Boolean map theory posits, that we can only select in the sense of conscious access and reportability one non-spatial feature like color at any given moment in time. Hence, our interpretation is that serial resource allocation arises from limits of conscious access to changing color values and not from limits of early sensory representation.